# Knowledge of the Serological Response to the Third BNT162b2 Vaccination May Influence Compliance of Healthcare Workers to Booster Dose

**DOI:** 10.3390/antib13030063

**Published:** 2024-08-01

**Authors:** Avi Magid, Khetam Hussein, Halima Dabaja-Younis, Moran Szwarcwort-Cohen, Ronit Almog, Michal Mekel, Avi Weissman, Gila Hyams, Vardit Gepstein, Netanel A. Horowitz, Hagar Cohen Saban, Jalal Tarabeia, Michael Halberthal, Yael Shachor-Meyouhas

**Affiliations:** 1Management, Rambam Health Care Campus, Haifa 3109601, Israel; khetamh@gmc.gov.il (K.H.); m_mekel@rambam.health.gov.il (M.M.); a_weissman@rambam.health.gov.il (A.W.); v_gepstein@rambam.health.gov.il (V.G.); n_horowitz@rambam.health.gov.il (N.A.H.); m_halberthal@rambam.health.gov.il (M.H.); y_shahor@rambam.health.gov.il (Y.S.-M.); 2School of Public Health, Ben-Gurion University of the Negev, Beer-Sheva 84105, Israel; 3The Ruth & Bruce Rappaport Faculty of Medicine, Technion—Israel Institute of Technology, Haifa 3200003, Israel; h_dabaja@rambam.health.gov.il; 4Infection Control Unit, Rambam Health Care Campus, Haifa 3109601, Israel; j_tarabeih@rambam.health.gov.il; 5Pediatric Infectious Disease Unit, Rambam Health Care Campus, Haifa 3109601, Israel; 6Virology Laboratory, Rambam Health Care Campus, Haifa 3109601, Israel; m_szwarcwort@rambam.health.gov.il; 7Epidemiology Unit, Rambam Health Care Campus, Haifa 3109601, Israel; r_almog@rambam.health.gov.il; 8Nursing Management, Rambam Health Care Campus, Haifa 3109601, Israel; ghyams@rambam.health.gov.il (G.H.); hagar.baruch@moh.gov.il (H.C.S.); 9Department of Pediatrics B, Rambam Health Care Campus, Haifa 3109601, Israel; 10Department of Hematology and Bone Marrow Transplantation, Rambam Health Care Campus, Haifa 3109601, Israel; 11Nursing Faculty, The Max Stern Yezreel Valley College, Yezreel Valley 1930600, Israel

**Keywords:** SARS-CoV-2, healthcare workers, vaccination, booster, compliance

## Abstract

Background: Previous studies showed that the fourth SARS-CoV-2 vaccine dose has a protective effect against infection, as well as against severe disease and death. This study aimed to examine whether knowledge of a high-level antibody after the third dose may reduce compliance to the fourth booster dose among healthcare workers (HCWs). Methods: We conducted a prospective cohort study among HCWs vaccinated with the first three doses at Rambam Healthcare Campus, a tertiary hospital in northern Israel. Participants underwent a serological test before the fourth booster vaccine was offered to all of them, with results provided to participants. The population was divided into two groups, namely those with antibodies below 955 AU/mL and those with 955 AU/mL and higher, a cutoff found protective in a previous study. Multiple logistic regression was carried out to compare the compliance to the fourth booster between the two groups, adjusted for demographic and clinical variables. Results: After adjusting for the confounding variables, the compliance was higher in those with antibody levels below 955 AU/mL (OR = 1.41, *p* = 0.05, 95% CI 1.10–1.96). In addition, male sex and age of 60 years and above were also associated with higher vaccination rates (OR = 2.28, *p* < 0.001, 95% CI 1.64–3.17), (OR = 1.14, *p* = 0.043, 95% CI 1.06–1.75), respectively. Conclusions: Knowledge of the antibody status may affect compliance with the booster dose. Considering waning immunity over time, reduced compliance may affect the protection of HCWs who declined the fourth dose.

## 1. Introduction

Three years after the beginning of the SARS-CoV-2 pandemic and despite the existence, safety, and effectiveness of the BNT162B2 vaccine, the SARS-CoV-2 continues to mutate and spread worldwide [1]. The third BNT162B2 dose (booster) vaccine was found to be effective in reducing the infection spread, severity of disease, and mortality rate during the Delta variant wave. In Israel, 90% of the adult population were vaccinated with three doses of the BNT162B2 vaccine by September 2021. However, when the Omicron variant began to circulate due to its high infectiousness, the infection spread in Israel was high despite the high vaccination rate. As a result, the Israeli Ministry of Health recommended the administration of the fourth BNT162B2 dose (booster) to populations exposed to higher risks, including adults aged 60 years and older, healthcare workers (HCWs), and immunocompromised adults. Since its introduction in December 2021, the Omicron variants have been circulating worldwide for more than 22 months, and there is evidence of weakened and waning immunity among vaccinated people, which also depends on the time since the last received dose [2,3,4,5,6,7]. There was also evidence that the BNT162B2 fourth (booster) vaccine dose has a protective effect, compared with those who received only the first two vaccine doses, but its effect is shorter than the third vaccine [8]. Therefore, vaccination appears to be the best way to protect the population and control the pandemic. The vaccination of HCWs is of major importance due to its potential to protect both HCWs and the patients [9] and due to HCWs’ high exposure to infected patients [10]. The term vaccine hesitancy is defined as “delay in acceptance or refusal of vaccination despite availability of vaccination services” [11]. According to previous studies, vaccine hesitancy is a common phenomenon worldwide [11]. Particularly, vaccine hesitancy also exists among HCWs, despite their advanced education in medical sciences and their clinical experience [12]. Previous studies showed that vaccine hesitancy among HCWs exists also with regard to SARS-CoV-2 vaccines, including the booster doses [13]. Factors associated with vaccine hesitancy among HCWs were previously studied; it was found that being a male, older age, and holding an M.D. degree was associated with a higher probability of SARS-CoV-2 vaccine acceptance. Moreover, a history of influenza, direct interaction with patients, and a higher perceived risk of infection with SARS-CoV-2 were also associated with a higher likelihood of SARS-CoV-2 vaccine acceptance [14].

A systematic review and meta-analysis that examined factors associated with vaccine acceptance by HCWs and included studies evaluating the acceptance of vaccines against influenza, pertussis, smallpox, anthrax, and hepatitis B found that desire for self-protection and desire to protect family and friends were strongly associated with vaccine acceptance [15]. A study which evaluated factors associated with influenza vaccine acceptance among healthcare workers in 171 German hospitals found that self-protection was the strongest factor associated with vaccine acceptance and that physicians had higher compliance with influenza vaccine than nurses [16].

Previous studies showed that antibody levels could predict the risk of being infected. Barda et al. found that a cutoff point of 700 BAU was protective against illness among individuals with pre-booster vaccination [17]. We have demonstrated that a cutoff point of 955 AU/mL (in a different method) was protective against illness [8]. Knowledge of their antibody levels may affect HCWs’ decisions as to whether or not they accept the booster vaccine dose. Concerns that may influence the HCWs’ decision not to receive the fourth dose may involve their familiarity with the vaccine’s adverse effects, which were partially known by this time. 

Our study aimed at assessing compliance with the fourth booster dose among HCWs who had knowledge about the antibody level for SARS-CoV-2 after the third vaccine dose.

## 2. Methods

We conducted a prospective cohort study among HCWs and retired employees vaccinated with the first two BNT162B2 doses at Rambam Healthcare Campus (RHCC), a tertiary 1000-bed hospital in northern Israel, which has 5520 employees, including 1220 physicians, 1880 nurses, 1137 paramedical workers, and 1283 administrative workers. The BNT162b2 vaccine was introduced to RHCC HCWs in January 2021, the same time it was introduced to all Israel’s HCWs. All RHCC HCWs who received at least two doses of the BNT162b2 vaccine and did not have a history of infection before the second dose were eligible to participate in the study upon their consent. Participants underwent serial serological tests at 1, 3, 6, 9, 12, and 18 months following the second vaccine dose (during February, April, July, and October 2021 and January and June 2022, respectively), with results provided to participants.

From this cohort, we took only those who received the third BNT162B2 booster vaccine, which was administered when the Delta variant was circulating, receiving a smaller cohort of 899 participants. The population was divided into two groups: those with antibody levels after the third dose below 955 AU/mL and those with 955 AU/mL and higher, based on a previous study which demonstrated the level as a protecting one [8].

In December 2021, a fourth (booster) vaccine dose was offered to individuals for whom four months had passed since their third vaccine dose. At this point in time, the Omicron variant began to circulate. The administration of the fourth vaccine was adjacent to the fourth serologic test time point (two months before) so that each individual knew his/her antibody levels when deciding whether to accept the fourth vaccine or not (Figure 1).

The study was approved by the hospital’s Internal Review Board (#021-021), and written informed consent was obtained from all participants.

## 3. Serology Assays

Serology testing was performed at 1, 3, 6, 9, 12, and 18 months post-second vaccine dose on LIAISON^®^ XL analyzer with the LIAISON SARS-CoV-2 TrimericS IgG assay (DiaSorin S.p.A., Saluggia, Italy) according to the manufacturer’s instructions. The method was described before [8]. Cut-off values for positive serology were 22 AU/mL, borderline 13–22 AU/mL; negative serology was reported for values <13 AU/mL. For values >799 AU/mL, serum was diluted on-board 1:20 with LIAISON TrimericS IgG diluent.

The serology tests at the 6th point were performed between 20 and 22 June 2022.

## 4. Statistical Analysis

Descriptive statistics, including mean, standard deviation, median, percentiles, counts, and percentages, were calculated for all the study variables. Differences between continuous variables were evaluated using a *t*-test. The exact two-tailed Wilcoxon matched-pairs signed rank test used abnormal distributions. Differences between categorical variables were evaluated using the chi-square test. Multiple logistic regression was carried out to compare the compliance to the fourth booster between the two groups (those with antibody levels after the third dose below 955 AU/mL, and those with 955 AU/mL and higher), adjusted for age, sex, smoking, Body Mass Index (BMI), flu-like disease, allergic response to one of the vaccines, and exposure to SARS-CoV-2 since the second vaccine and since the last vaccine. Odds Ratio (OR), 95% Confidence Interval (CI), and *p*-value were presented.

For each result, *p*-value < 0.05 was considered statistically significant. SPSS version 28 was used for all statistical analyses.

## 5. Results

### 5.1. Description of the Study Population

Out of 899 of the cohort participants, only 232 (25.8%) chose to receive the fourth vaccine dose. The mean serology level was statistically significantly higher in those who did not receive the fourth vaccine compared to those who received the fourth vaccine (Table 1). The rate of vaccination with the fourth vaccine was statistically significantly higher in those who were positive for SARS-CoV-2 since the second vaccine compared to those who were not (Table 1).

### 5.2. Multiple Logistic Regression to Compare the Compliance to the 4th Booster between the Two Groups

After adjusting for sex, age, smoking, BMI, having flu-like disease, having allergic response to previous vaccines, and exposure to SARS-CoV-2 positive people since the second vaccine and since the last vaccine, the compliance to the fourth booster was higher in those with antibody levels below 955 AU/mL compared to those with antibody level of 955 AU/mL and above OR = 1.41, *p* = 0.05, 95% CI 1.10–1.96) (Table 2). In addition, male sex and age of 60 years and above were also associated with higher vaccination rate (OR = 2.28, *p* < 0.001, 95% CI 1.64–3.17), (OR = 1.14, *p* = 0.043, 95% CI 1.06–1.75), respectively (Table 2). No association was found between smoking, having flu-like disease, having an allergic response to previous vaccines, and exposure to SARS-CoV-2 since the second vaccine and since the last vaccine and the fourth vaccine acceptance.

## 6. Discussion

In this study, we aimed to assess the association between knowledge of the antibody levels before the fourth vaccine dose and the acceptance of the fourth booster dose.

We have found that compliance was lower among those who knew their antibody level was higher compared with those who knew their antibody level was lower. Our results showed that the acceptance rate of those who knew their antibody level was higher was lower by 41% than the acceptance rate of those who knew their antibody level was lower. This is a unique observation.

The level of the antibodies, whether neutralizing or other, may reflect the risk for infections. It was shown before, by us and others, that the presence of antibodies may protect from SARS-CoV-2 infection [8,18,19]. In other diseases, the presence of antibodies by either previous infection or vaccination is considered adequate for protection from the disease. The method of testing the antibody level and deciding whether vaccination is required is used among healthcare workers, for example, in the case of measles, in which healthcare workers with a certain threshold antibody level and above are considered vaccinated and do not need to get vaccinated [20]. There are some cases in which antibody level and the use of cutoff values are crucial for protection, such as in the case of the hepatitis B vaccine [21], whereas in other diseases, such as flu or other respiratory viruses, the ability to test for the antibodies is not a daily practice even after disease or vaccination, i.e., it is not a daily practice to test the antibody level for influenza or other respiratory viral infections before vaccination.

In the case of SARS-CoV-2, ongoing publication and data suggested the use of serology testing for some purposes, the determination of whether a non-vaccinated person is immune by previous presumed infection and also in cases of immunocompromised patients that were vaccinated but are not clearly immune among them [22]. With the knowledge of a protection level (and maybe validation in the future), a person with a severe reaction to the vaccine can take a serology test in order to understand the immunity by previous vaccination and consider the need for a booster dose. However, in the general population, unlike diseases such as measles and hepatitis B, in the case of SARS-CoV-2, even if one has antibodies, one should receive the vaccine. This is partly due to the waning effect of the BNT162b2 vaccine [8]. In other words, the knowledge of the antibody levels may be useful for special cases to decide whether to accept the BNT162b2 vaccine but not for the general population. Naturally, HCWs, who are more vulnerable due to their exposure to patients, may not make their decision whether to accept the vaccine based on their antibody level. Our results show that knowledge of the antibody level may affect their decision and, therefore, may expose them to higher risk.

In late 2020, when the vaccine for SARS-CoV-2 was introduced [23], HCWs in our institution were offered a serology test since many of them were hesitating before vaccination and had the assumption that they were already positive. Offering the opportunity was accepted by 82.5% of them, and most of them were negative and, therefore, were able to receive the vaccine with almost 100% compliance among the staff. In other words, when the SARS-CoV-2 vaccine was completely new, the hesitancy among healthcare workers in our institution was high, and knowing their negative serology test results might have increased their vaccine acceptance [24]. In this study, we tested the acceptance of the booster fourth SARS-CoV-2 vaccine among healthcare workers as a function of their knowledge of their serology test results reflecting the third vaccine and demonstrated that the acceptance of the booster vaccine by healthcare workers was also associated and may be affected by the knowledge of the serology level. In this case, unlike the first serology tests, after the third vaccine, there were more positive serology cases, and our results show that higher antibody levels may be a contributor to BNT162b2 vaccine hesitancy among HCWs.

Vaccine hesitancy among healthcare workers is of major interest and concern all over the world [25], and in the past 2 years, the SARS-CoV-2 vaccine acceptance was studied in many countries [26,27]. In our current study, higher antibody levels were also related to a reduction in vaccination compared with lower levels, and we also found that males were more likely to receive the fourth vaccine dose regardless of their antibody level. This observation is consistent with other studies which were carried out among HCWs or the general population, and found that males were more likely to accept vaccines [26,27,28]. Among women, especially during the childbearing age, there is a decrease in vaccination acceptance and concern about vaccination safety. In a study conducted by Biswas et al. in 2021, reviewing 35 studies (76,471 people) regarding HCWs vaccination for SARS-CoV-2 from all over the world, male sex was an enabling factor in more than 70% of the studies (25 out of 35 studies) [14]. In another study by Farah et al., which was carried out in the general population of the USA, it was also found that male sex was related to higher vaccine acceptance in the general population [26].

Our data also demonstrated differences in age as a contributor to accepting the fourth booster dose, where those aged 60 years and above were more likely to accept the fourth booster vaccine, independent of their serology test result. Previous studies showed that those aged 60 years and above are at higher risk for severe SARS-CoV-2 infections, including hospitalization, severe morbidity, and mortality, and that the vaccine was effective for them [29]. It was also shown by us and others that antibody levels may be, in some cases, lower among those who are >60 years old [30].

Many studies demonstrated higher vaccine coverage and higher rates of booster coverage among those above 60 years in the general population as well as among HCWs [31]. The use of boosters was also proven to prevent deaths and hospitalizations among the elderly [18]. Our results with regard to the association between older age and the acceptance of the booster vaccine are consistent with the previous literature.

Preparedness for the next pandemic by learning lessons from the SARS-CoV-2 pandemic is a major goal of health organizations worldwide [32]. Although the measures which were taken during the pandemic were not innovative, such as physical distancing, mask-wearing, and quarantines, the pandemic was considered an exceptional international crisis and led to the usage of the term “the new normal”, which is defined as “life at a constant risk which can only be contained and regulated but cannot be totally overcome” [33]. The new normal refers to the changes caused by the SARS-CoV-2 pandemic. The new normal means that we should live in a constant state of preparedness [33]. The new normal is continuously challenging healthcare systems, especially when variants of Omicron are rapidly emerging, together with other emerging threats such as polio and seasonal influenza, which are all in the background of the post-pandemic fatigue of healthcare workers [34]. This rapid emergence of Omicron variants, together with the waning immunity of the vaccines shown in previous works, emphasizes the necessity of constant development of booster doses and compliance with the booster doses by the general public and, particularly, by healthcare workers [34]. Increasing the compliance of the general population, as well as HCWs, to accept recommended vaccines is crucial for controlling the pandemic [32]. Understanding the broad factors associated with vaccine acceptance, which include mapping specific population groups with special characteristics, and understanding the factors associated with vaccine acceptance within these groups is crucial to living in the new normal era [35]. We found that knowing the antibody level may affect HCWs’ decisions as to whether or not they will receive the booster vaccine. This finding sheds light on understanding factors associated with the vaccine hesitancy of HCWs and may be taken into consideration by decision-makers when deciding whether to offer, in specific circumstances, a serology test to HCWs.

Our study also coincides with a theoretical model.

The Health Belief Model (HBM) associates beliefs, perceptions, and health behavior, such as a healthy lifestyle, injury prevention, compliance with screening tests, and vaccine acceptance [36]. According to the HBM, the individual’s decision whether to perform a specific health behavior related to a specific disease depends on the individual’s perception of the disease threat, the individual’s perception of the usefulness of performing this health behavior (the perceived benefits of performing the behavior), and the individual’s barriers for performing the behavior [37]. Some previous studies explored the association between components of the HBM and SARS-CoV-2 vaccine acceptance. An Israeli study found that parents’ decision whether to vaccinate their adolescent children against SARS-CoV-2 was associated with the parents’ perceived SARS-CoV-2 threat and that this association was mediated by the SARS-CoV-2 vaccine benefits perceived by parents. This study highlighted the contribution of the HBM to explaining the acceptance of vaccine against SARS-CoV-2 [37]. In a systematic review examining the association between HBM components and SARS-CoV-2 vaccine hesitancy, it was found that the perceived severity of the disease, perceived susceptibility, cues to action, perceived benefits, and self-efficacy were negatively associated with vaccine hesitancy. Moreover, perceived barriers were found to be positively associated with vaccine hesitancy [38]. Establishing the HBM as a theoretical framework for understanding factors associated with SARS-CoV-2 hesitancy may be essential to developing tailored and targeted interventions to reduce hesitancy [38]. The findings of our study may also be explained by the HBM. At the beginning of the SARS-CoV-2 pandemic, the perceived SARS-CoV-2 threat was high among the general population, as well as among HCWs (our study population). Hence, when the first three vaccines were introduced and offered to HCWs, given the high perceived threat and the perceived benefits (for example, future ability to enter public places), vaccination compliance was high, conforming to the HBM. However, when the fourth booster was offered, HCWs were already vaccinated with three vaccines against SARS-CoV-2, and the Omicron variant, which was circulating in this period, was perceived as a less severe disease than previous variants, and therefore, the perceived threat was expected to decrease, and, according to the HBM, the compliance to the booster vaccine was also expected to decrease. The association between knowledge of high antibody levels detected by serology tests and lower acceptance of the booster dose established in our study may also be explained by the perceived threat component of the HBM, which may be affected by the serology test results.

## 7. Limitations

First—The study is a single-center study, where all of the study participants were HCWs with more health-related knowledge compared to the general population and with a better ability to search for information regarding their antibody level.

Second—there are no data regarding neutralizing antibodies in our study, but there are data regarding the correlation between them in other studies in the past [35].

Third—We did not ask participants about their decisions regarding receiving the fourth booster dose, and although we found an association between antibody levels and vaccine acceptance, we cannot assume that the reason for accepting or not accepting the vaccine was the antibody level but rather an observation that was found. However, it is reasonable to think that at least part of the participants, which were HCWs, knew their serology test results and whether they were considered high before accepting or not accepting the vaccine.

## 8. Conclusions

Our study demonstrated the possible influence of the knowledge of antibody level on receiving the fourth BNT162b2 vaccine booster dose for SARS-CoV-2 among HCWs in a tertiary hospital. Our finding coincides with the HBM. Other known factors, such as age and male sex, were also found as enabling factors. Considering waning immunity over time, reduced compliance may affect the protection of HCWs who declined the fourth dose. Our findings may be helpful for healthcare decision-makers, especially in cases when faced with hesitancy and fear of side effects.

## Figures and Tables

**Figure 1 antibodies-13-00063-f001:**
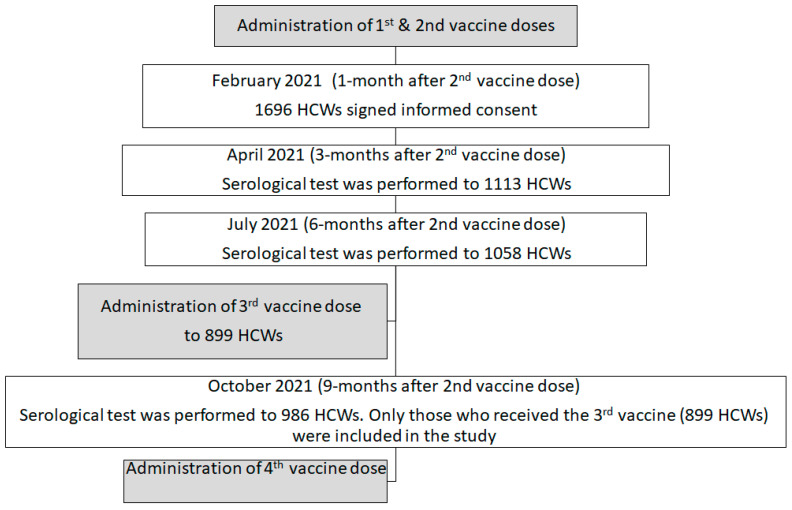
The serology cohort description based on our previous study on the same cohort [8].

**Table 1 antibodies-13-00063-t001:** Characteristics of participants who were vaccinated with the 4th vaccine and participants who were not vaccinated with the 4th vaccine. Comparison of continuous variables was conducted using *t*-test. Comparison between categorical variables was conducted using Chi-square test.

Variable	Vaccinated with the 4th Vaccine	Not Vaccinated with the 4th Vaccine	*p*
N	232 (25.8%)	667 (74.2%)	--
Mean age in years (Median, IQR, min–max)	54.96 (56, 16, 28–84)	61.85 (49, 17, 19–87)	0.615
Sex (Male)	92 (39.6%)	142 (21.3%)	<0.001
Mean serology level (Median, IQR)	1983.11 (1360, 1583)	2444.73 (1630, 2215)	0.004
Mean BMI (Median, IQR)	26.30 (26.9, 7.1)	26.90 (25.5, 7.5)	0.215
Smoking	18 (7.8%)	76 (11.4%)	0.110
Flu-like disease	23 (9.9%)	68 (10.2%)	0.9
Allergic reaction to one of the vaccines	0 (0%)	27 (4.0%)	<0.001
Exposed to SARS-CoV-2 since last vaccine	54 (23.2%)	198 (29.6%)	0.07
Positive for SARS-CoV-2 since second vaccine	3 (1.3%)	25 (3.7%)	0.05

**Table 2 antibodies-13-00063-t002:** Multiple logistic regression to compare the compliance to the fourth booster between the two groups.

Variables	OR	95% CI	*p*
Serology			
>955 AU/mL	1	Ref.	
<955 AU/mL	1.41	1.10–1.96	0.05
Sex			
Female	1	Ref.	
Male	2.28	1.64–3.17	<0.001
Age (years)			
<60	1	Ref.	
60+	1.14	1.06–1.75	0.008
Smoking			
Smoker	1	Ref.	
Non-smoker	1.64	0.94–2.89	0.084
BMI			
>30	1	Ref.	
25–30	0.84	0.56–1.27	0.417
<24.9	0.60	0.40–0.89	0.470
Flu-like disease			
Yes	1	Ref.	
No	0.77	0.45–1.31	0.331
Allergic reaction to one of the vaccines			
Yes	1	Ref.	
No	1.48	0.98–2.58	0.998
Positive for SARS-CoV-2 since last vaccine			
Yes	1	Ref.	
No	1.09	0.76–1.58	0.637
SARS-CoV-2 since second vaccine			
Yes	1	Ref.	
No	2.49	0.56–11.07	0.229

## Data Availability

The data are not publicly available due to privacy and ethical restrictions.

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
