# Peer review of "Knowledge of the Serological Response to the Third BNT162b2 Vaccination May Influence Compliance of Healthcare Workers to Booster Dose"

_2073-4468, 2024, doi:10.3390/antib13030063_

Round 1

Reviewer 1 Report

Comments and Suggestions for Authors

The paper proves the obvious thesis of the influence of the realised potential threat on protective action, which stems from the psychic construction of homo sapiens and has been confirmed repeatedly in many areas. It therefore has little novelty value.

Most people during the COVID pandemic, especially from the medical community, based their decision to receive further doses of vaccine on their immunisation status.

Concerns (real or otherwise) that influenced the decision not to receive the 4th dose of vaccine were not addressed in the paper.

In the 'materials and methods', please detail the chapters: study design and characteristics of the study population.

No characterisation of the vaccines used and their efficacy against viral variants.

No indication of the period during which the study was conducted and no data on the epidemiological situation of the circulating variants at that time.

Please provide a graphic (timeline) to give a quick overview of the structure and progress of the study.

Under "Statistical Analysis", please specify which statistical tests were used and for what purpose. How was the minimum size of the study groups calculated in order to obtain reliable strength of conclusions.

Line 122-124 - Table 1 does not present these data. Please expand the table.

Table 2 - no description of designations: OR, CI and p.

Please provide a comparison between groups using appropriately selected statistical methods and provide complete data from these comparisons.

"Discussion" has many repetitions of the same aspects and several irrelevant digressive threads. It needs to be rewritten and structured for readability.

Author Response

Dear Reviewer,

Thank you for reviewing our paper and for your important comments. We made major changes to our paper based on your comments and recommendations. Following is our answer to each of your comments:

The paper proves the obvious thesis of the influence of the realised potential threat on protective action, which stems from the psychic construction of homo sapiens and has been confirmed repeatedly in many areas. It therefore has little novelty value.

Most people during the COVID pandemic, especially from the medical community, based their decision to receive further doses of vaccine on their immunisation status.

Answer: We agree with the reviewer that this kind of issue has been addressed in other circumstances, but not specifically for the COVID vaccines, which is a unique situation.

In COVID, unlike measles, even if you have antibodies, you should receive the vaccine, partially due to its waning effect. We added a paragraph to the discussion in order to clarify it.

Concerns (real or otherwise) that influenced the decision not to receive the 4th dose of vaccine were not addressed in the paper.

Answer: The cohort participants were not asked about their possible concerns regarding their decision to receive the 4th vaccine. However, we added to the “introduction” chapter a possible concern – their familiarity with the vaccine’s adverse effects, which were partially established by this time.

In the 'materials and methods', please detail the chapters: study design and characteristics of the study population.

Answer: The “Methods” chapter was updated to include the study design, as well as the characteristics of the study population

No characterisation of the vaccines used and their efficacy against viral variants.

Answer: Thank you for this important comment, description of the characterization of the vaccines used, as well as their efficacy against the different variants was added to the “Introduction” chapter. A description of the specific type of vaccines used was also added to the “Methods” chapter. We emphasis that the only vaccine used in Rambam Health Care Center is the BNT162b2.

No indication of the period during which the study was conducted and no data on the epidemiological situation of the circulating variants at that time.

Answer: A description of the study period, as well as the epidemiological situation of the circulating variants at that time were added to the “methods” chapter.

Please provide a graphic (timeline) to give a quick overview of the structure and progress of the study.

Answer: A graphic timeline was provided (Figure 1).

Under "Statistical Analysis", please specify which statistical tests were used and for what purpose. How was the minimum size of the study groups calculated in order to obtain reliable strength of conclusions.

Answer: The statistical tests which were used, including the statistical tests which were used in the updated Table 1 were specified in the “Methods” chapter (under “Statistical Analysis”).

Line 122-124 - Table 1 does not present these data. Please expand the table.

Answer: Table 1 was updated to include comparison of data between participants who where vaccinated with the 4th vaccine and participants who were not vaccinated with the 4th vaccine (the study’s main outcome variable). Results description (which refers to Table 1) was updated accordingly.

Table 2 - no description of designations: OR, CI and p.

Answer: The full terms were given in the text (in the “Methods” chapter).

Please provide a comparison between groups using appropriately selected statistical methods and provide complete data from these comparisons.

Answer: Comparison was provided in the updated Table 1, as well as in the “Results” chapter. Appropriate statistical methods were selected and described in the “Methods” chapter (under “Statistical Analysis”).

"Discussion" has many repetitions of the same aspects and several irrelevant digressive threads. It needs to be rewritten and structured for readability.

Answer: The discussion was corrected. Some paragraphs and sentences were added to achieve a more structured and readable text and to avoid digression.

Reviewer 2 Report

Comments and Suggestions for Authors

Dear authors,

Your manuscript “Knowledge of the serological response to the 3rd BNT162b2 vaccine may influence compliance of health care workers to booster dose” describes the assessing the compliance with the 4th booster dose among healthcare workers and the connection with different factors. The manuscript contains all necessary parts, well written and gives interesting data.

However, some issues should be revised.

First, my suggestion is to make change the name of the manuscript, and change “response to the 3rd BNT162b2 vaccine” to “response to the 3rd BNT162b2 vaccination”.

Second, some remarks to be solved:

Introduction section - You discuss vaccination and vaccine boosters, but do not describe for which vaccines these data were obtained; however, there are many vaccines on the market with different principles of action, and the data provided in the introduction may be true for all of them. I think it's worth clarifying this issue here. The same issue is in Methods section.

L115 – BMI – the full term should be given here (for the first time).

L145 – what is “OR”?

Table 2 contains blank table cells. Why?

Author Response

Dear Reviewer,

Thank you for reviewing our paper and for your important comments. We made major changes to our paper based on your comments and recommendations. Following is our answer to each of your comments:

Your manuscript “Knowledge of the serological response to the 3rd BNT162b2 vaccine may influence compliance of health care workers to booster dose” describes the assessing the compliance with the 4th booster dose among healthcare workers and the connection with different factors. The manuscript contains all necessary parts, well written and gives interesting data.

However, some issues should be revised.

First, my suggestion is to make change the name of the manuscript, and change “response to the 3rd BNT162b2 vaccine” to “response to the 3rd BNT162b2 vaccination”.

Answer: Thank you for your suggestion. We changed the name of the manuscript accordingly.

Second, some remarks to be solved:

Introduction section - You discuss vaccination and vaccine boosters, but do not describe for which vaccines these data were obtained; however, there are many vaccines on the market with different principles of action, and the data provided in the introduction may be true for all of them. I think it's worth clarifying this issue here. The same issue is in Methods section.

Answer: Thank you for this important comment. We added an explicit description of the specific vaccines for which our data were obtained in the “introduction” chapter, as well as in the “methods” chapter. We emphasis that the only vaccine used in Rambam Health Care Center is the BNT162b2.

L115 – BMI – the full term should be given here (for the first time).

Answer: The full term was given in the text

L145 – what is “OR”?

Answer: “OR” is Odds Ratio. The full term was given in the text (in the “Methods” chapter).

Table 2 contains blank table cells. Why?

Answer: Table 2 contains blank cells in rows representing reference categories, and, as such, P-value and Confidence interval are not applicable. We merged some of the cells and mentioned “Ref.” in other empty cells such that no empty cells do not exist in the table.

Round 2

Reviewer 1 Report

Comments and Suggestions for Authors

Some of the comments have been sufficiently taken into account, but the article still has important shortcomings that need to be corrected:

Fig. 1 is illegible. Please correct the figure so that it is understandable and legible. You can use larger font size and colors.

Table 1 is not signed.

Table 1 has an inappropriate format. Please use the one adopted by the publishing house (in terms of lines).

Table 1 - the arithmetic mean of "Mean age in years", "Mean serology level", "Mean BMI" was used.

Arithmetic averages are misleading because they are sensitive to extreme values.

In the first case, providing Median/Mean (IQR b, min–max) has greater substantive value, and in the second case, Geometric Mean (Geo SD) or Median (IQR), and in the third case, BMI - Median (IQR) and using appropriate tests for comparisons between groups, e.g. p-Value using an exact two-tailed Wilcoxon matched-pairs signed rank test for an abnormal distribution or Mann–Whitney test between cohorts.

Table 2 has an inappropriate format. Please use the one adopted by the publishing house (in terms of lines).

Author Response

Dear reviewer:

Thank you for reviewing our corrected version of the manuscript. The manuscript was corrected in accordance with your comments.

Fig. 1 is illegible. Please correct the figure so that it is understandable and legible. You can use larger font size and colors.

 Answer: Figure 1 was corrected. Larger font size was used, and the entire figure was enlarged.

Table 1 is not signed.

Answer: Table 1 was corrected to be signed

Table 1 has an inappropriate format. Please use the one adopted by the publishing house (in terms of lines).

 Answer: Table 1 was corrected to have the appropriate format.

Table 1 - the arithmetic mean of "Mean age in years", "Mean serology level", "Mean BMI" was used.

Arithmetic averages are misleading because they are sensitive to extreme values.

In the first case, providing Median/Mean (IQR b, min–max) has greater substantive value, and in the second case, Geometric Mean (Geo SD) or Median (IQR), and in the third case, BMI - Median (IQR) and using appropriate tests for comparisons between groups, e.g. p-Value using an exact two-tailed Wilcoxon matched-pairs signed rank test for an abnormal distribution or Mann–Whitney test between cohorts.

Answer: Thank you for this important comment. Medians, IQRs were evaluated and added to Table 1 for the "Mean age in years", "Mean serology level", "Mean BMI" variables. Statistical tests were updated accordingly. The “Method” chapter was also updated accordingly.

Table 2 has an inappropriate format. Please use the one adopted by the publishing house (in terms of lines).

Answer: Table 2 was corrected to have the appropriate format

Reviewer 2 Report

Comments and Suggestions for Authors

Dear authors,

Thank you for the revision made, currently I have no remarks.